# In-hospital real-time prediction of COVID-19 severity regardless of disease phase using electronic health records

**Hyungjun Park**[1], **Chang-Min Choi**[2,3], **Sung-Hoon Kim**[4], **Su Hwan Kim**[5,6], **Deog Kyoem Kim**[5,7], **Ji Bong Jeong**[5,6]*

**1** Division of pulmonology and Critical Care Medicine, Department of Internal Medicine, Gumdan top hospital, Incheon, South Korea, **2** Division of Pulmonology and Critical Care Medicine, Department of Internal Medicine, Asan Medical Center, University of Ulsan College of Medicine, Seoul, South Korea, **3** Division of Oncology, Department of Internal Medicine, University of Ulsan College of Medicine, Asan Medical Center, Seoul, South Korea, **4** Department of Anesthesiology and Pain Medicine, Asan Medical Center, Ulsan College of Medicine, Seoul, South Korea, **5** Department of Internal Medicine, Seoul National University College of Medicine, Seoul, South Korea, **6** Division of Gastroenterology, Department of Internal Medicine, Seoul Metropolitan Government Seoul National University Boramae Medical Center, Seoul, South Korea, **7** Division of Pulmonary and Critical Care Medicine, Department of Internal Medicine, Seoul Metropolitan Government Seoul National University Boramae Medical Center, Seoul, South Korea

* jibjeong@snu.ac.kr

**Data Availability Statement:** The data that support the findings of this study were provided by the Institutional Review Board (IRB) of Boramae Medical Center. While these data are not publicly

## Abstract

Coronavirus disease 2019 (COVID-19) has strained healthcare systems worldwide. Predicting COVID-19 severity could optimize resource allocation, like oxygen devices and intensive care. If machine learning model could forecast the severity of COVID-19 patients, hospital resource allocation would be more comfortable. This study evaluated machine learning models using electronic records from 3,996 COVID-19 patients to forecast mild, moderate, or severe disease up to 2 days in advance. A deep neural network (DNN) model achieved 91.8% accuracy, 0.96 AUROC, and 0.90 AUPRC for 2-day predictions, regardless of disease phase. Tree-based models like random forest achieved slightly better metrics (random forest: 94.1% of accuracy, 0.98 AUROC, 0.95 AUPRC; Gradient boost: 94.1% of accuracy, 0.98 AUROC, 0.94 AUPRC), prioritizing treatment factors like steroid use. However, the DNN relied more on fixed patient factors like demographics and symptoms in aspect to SHAP value importance. Since treatment patterns vary between hospitals, the DNN may be more generalizable than tree-based models (random forest, gradient boost model). The results demonstrate accurate short-term forecasting of COVID-19 severity using routine clinical data. DNN models may balance predictive performance and generalizability better than other methods. Severity predictions by machine learning model could facilitate resource planning, like ICU arrangement and oxygen devices.

## Introduction

The COVID-19 infection swiftly propagated on a global scale, leading the World Health Organization (WHO) to officially declare it as a pandemic [1]. Although more than half of all infections are subclinical in young ages, 43% of older than 65 get hospitalized and 76% were died

accessible, they can be provided upon a reasonable request and with the approval of the IRB. For further inquiries or data requests, please directly contact the IRB of Boramae Medical Center at Tel: 82-2-870-1851. We have acknowledged the concern about the authors being the sole individuals responsible for data access and have taken appropriate measures to address this.

**Funding:** This work was supported by research funding from the Seoul Metropolitan Government Seoul National University (SMG-SNU) Boramae Medical Center (04-2022-0004), and a grant of the Korea Health Technology R&D Project through the Korea Health Industry Development Institute (KHIDI), which is funded by the Ministry of Health & Welfare, Republic of Korea (grant number: HI18C2383) The funders had no role in study design, data collection and analysis, decision to publish, or preparation of the manuscript.

**Competing interests:** The authors have declared that no competing interests exist.

until September 2021 [2]. To control the spread of COVID-19 infection, pharmaceutical (vaccination) [3] and non-pharmaceutical interventions such as using face masks, isolation, quarantining, and social distancing have been exercised [4, 5]. Although the difference of stringency and period of covid-19 restriction, most of the countries conducted severe restriction more than 10 months [4]. At the peak of the pandemic, these interventions were necessary to reduce shortages in hospital care resources, particularly among critical care resources such as intensive care units (ICUs) and oxygen supplies [6, 7]. As a result of the number of COVID-19 patients exceeding the hospital's capacity, some patients received inadequate care, resulting in poorer results than expected for the natural course of COVID-19 [6]. Moreover, severe COVID-19 patients require high-flow oxygen devices and mechanical ventilation, and preparation of those facilities was warranted [8]. Therefore, the Korean government replaced public hospitals with infectious disease-dedicated hospitals, monitored the daily number of severe COVID-19 patients, and determined whether this number would exceed the hospital capacity [9].

Prediction of future need for intensive care is required for controlling capacities for facilities in medical centers. Timely preparation of essential medical devices based on short-term forecasts (1 day or 2 days) is critical for ensuring efficient management of healthcare resources. By predicting short-term patient outcomes, informed decisions can be made regarding the potential transfer of patients to alternate hospitals or care units within the same facility's ICU. With the development of artificial intelligence, several prediction models were established to predict the short-term outcomes of COVID-19 [10–12]. In the early phase of the pandemic, many models were developed using static predictions—these predicted future events at admission [10–12], but did not indicate when the event would occur [13]. In contrast to static predictions, dynamic predictions or real-time predictions estimate a temporal event, such as the daily risk of acute kidney injury [14] or need for ICU care [15]. Dynamic prediction in the medical context pertains to the daily forecasting of impending outcomes, such as complications or mortality. Concerning COVID-19, the near-future possible outcome include patient mortality [11], the requirement for mechanical ventilation [16], as well as discharge or mortality [17]. The predictive capacity for complications or mortality empowers healthcare workers to concentrate on addressing correctable issues in patients and to adequately prepare for future intensive care unit (ICU) demands related to mechanical ventilation. Given the restrictions on available resources, forecasting not only adverse outcomes but also predicting the potential recovery of patients from mechanical ventilation or oxygen support assumes importance in hospital resource management. Consequently, making accurate predictions about patients' potential recovery from mechanical ventilation and their subsequent discharge from the ICU enables informed decision-making concerning the transfer of other patients in need of mechanical ventilation. However, previous studies focused only on the start date of the event and not on recovery.

We examined various machine learning and deep learning models to predict the daily severity of COVID-19 at a hospital dedicated to COVID-19 patients. In addition, we assessed the importance of input variables in each model and considered the possible hazard of bias associated with models that displayed exceptional performance.

## Material and methods

### Clinical dataset description

Deidentified patients with a confirmed diagnosis of COVID-19 from January 1, 2020, to October 31, 2021 at Boramae Medical Center (BMC) were consecutively included. The authors obtained access to this data set from February 1, 2022, following approval from the

Institutional Review Board (IRB). BMC was dedicated to infectious diseases and only admitted COVID-19 patients. During the pandemic, BMC used telemonitoring to monitor patients with COVID-19 in residential treatment centers. In case of symptom progression at a residential treatment center, the patient was referred to BMC. All patients identified with COVID-19 were diagnosed by nasopharyngeal swab reverse transcription-polymerase chain reaction. Patients younger than 18 years of age were excluded from this study. This study was approved by the ethics committee of the BMC and was conducted as per the Declaration of Helsinki. The requirement for informed consent was waived by the ethics committee of the BMC (approval number 10-2021-130), considering the retrospective nature of the study.

## COVID-19 severity as the target outcome

The disease severity defined by World Health Organization (WHO) clinical progression scales, was recorded daily at BMC at midnight [18]. The primary objective of our study was to forecast the severity of the condition prior to the event, focusing on predictions for day 0, 1, and 2. It was deemed sufficient to forecast the patient's severity up to 2 days in advance to adequately prepare the necessary oxygen and ICU resources. The WHO clinical progression scale ranges from 0 (not infected) to 10 (dead) [18]. In our model, the score was divided into 3 scores, i.e., mild cases (monitored, no oxygen therapy; WHO scale < 5), moderate disease (hospitalized, need oxygen by mask or nasal prong; WHO scale = 5), and severe disease (oxygen by high flow, non-invasive ventilation, or mechanical ventilation; WHO scale > 5). Utilization of the oxygen device was meticulously documented with corresponding timestamps, thereby ensuring that all daily fluctuations in severity were accurately captured on an daily basis.

## Data preprocessing and feature selection

From the electronic medical record system at BMC, we extracted the following data: demographic features (age, sex, underlying disease, smoking status), daily symptom records, laboratory data, drug use, vital signs. (S1 and S2 Tables) The laboratory data encompassed a range of parameters, including complete blood cell count, chemistry markers (such as protein, albumin, liver function tests, BUN/Creatinine), electrolyte levels, inflammatory markers (CRP, LDH, ferritin, procalcitonin), and coagulation measurements. The selection of these specific variables was determined by the attending clinicians responsible for patient care, who deemed them relevant for the management and treatment of COVID-19 patients. All these data served as input variables utilized in the prediction of daily severity fluctuations.

The binary and categorical features such as the presence of symptoms and type of oxygen device used were categorized as 0 or 1 by one-hot encoding. Although information on the type of oxygen device and fraction of oxygen was available, it was not included in the model to prevent data leakage because it directly represented the outcome. Antiviral agent use was treated as a binary variable and steroid use, which was represented as daily summed doses, was treated as a continuous variable. For continuous variables, data cleansing was conducted by removing error values (such as negative blood pressure values) and capping outlier values. Data clipping was usually conducted in the 1st and 99th percentile values [15]; however, we used outlier values decided by the clinicians as the data clipping points because several 1st and 99th percentile values did not represent severity of the laboratory values. Each data point is described in S1 Table. Among the extracted features, rarely checked laboratory examinations or vital signs, such as HbA1c and neurologic examinations, were excluded from our input dataset. Patients' symptoms were recorded daily and categorized into 18 types. Positive symptoms during admission are described in S2 Table. Every day, all daily symptom records were scrutinized and marked either as present or absent. If a patient was intubated, the corresponding symptom was deemed to be absent.

## Data exclusion and imputation

Data integrity is important for precise prediction [19]. Laboratory test were less conducted than vital sign data because patients with mild COVID-19 infection did not frequently undergo laboratory tests. The admission events were excluded if >50% of their vital signs data, >80% of their laboratory data, or their COVID-19 severity outcome data were missing during the entire admission period (S1 Fig).

Missing value imputation was conducted in two steps. First, if the variable had an existing previous value, the last observation carried forward method was used (such as laboratory data and vital signs). Second, when the patient was hospitalized without laboratory results, we assumed that the prediction based on normal values until abnormal data were observed would be robust. As many data distributions were skewed, the mode was appropriate for representing normal values in our dataset. (S2 and S3 Figs) Thus, no previously observed data were imputed with each mode value.

## Models

We evaluated several models for predicting the daily severity of COVID-19, classifying each model as a non-temporal or temporal model. The non-temporal model that received daily data consisted of logistic regression, random forest [20], gradient boost (XG boost) [21], and deep neural networks (DNN) [22]. The detailed hyperparameters of the models were described in Supplement methods. On the day of prediction, the model receives information regarding vital signs in the last 8 hours (16–24 hour) and laboratory and symptom data along with daily information. As temporal model, Transformer received trend information for prediction [23]. The model receives input as trend data, thus, the vital sign trend of n-day and lab data of n-day. The vital sign and laboratory data had different frequencies; the index time window of the vital sign was defined as 8 hours, and that of the lab was 24 hours. The detailed input pipeline is explained in Supplemental methods. (S1 File) We evaluated the model using different input lengths by changing it every two hospital days to determine its effect on the model.

The source code used for developing our model and for drawing the figure was implemented by Python (version 3.8), numpy (version 1.22), scikit-learn (version 1.0.2), pandas (version 1.4.0), Pytorch (version 1.10), and matplotlib (version 3.5.1).

## Statistical analysis

The baseline characteristics were described as mean and standard deviation for continuous variables and number and percentage for binary variables. The difference between continuous variables was calculated by ANOVA; binary variables were calculated using the chi-square test. Model performance was evaluated by area under the receiver operating curve (AUROC) and area under the precision-recall curve (AUPRC), which involved multi-class classification. AUROC and AUPRC were calculated by binarization to the target and the rest of the target. Each target severity was plotted by each receiver operating characteristic curve and precision-recall curve. For a more comprehensive comparison of the models, we implemented a decision curve analysis [24]. Given that this methodology was originally conceptualized for binary classification, it necessitated the modification of the data format to a "one versus the rest" approach for our predictions classified as mild, moderate, or severe. The patients were divided into training (60%), validation (20%), and test (20%) sets by stratified splitting, and the model was trained based on the validation set using early stopping. To calculate the mean and standard deviation of performance metrics according to the input length, we repeated splitting and training five times per input length.

The model importance was computed using Shapley additive explanation (SHAP) values, which is model-independent. Using this strategy, each of the random forest, gradient boost, and deep neural network models assessed the importance of the variable. Input variable importance was assessed based on the data type, distinguishing between hospital/treatment-dependent factors (extrinsic factors) and patient-dependent factors (intrinsic factors). Hospital/treatment-dependent factors encompassed the number of vital sign occurrences (count, variance) and the dosage of drugs (max, min, median). On the other hand, patient-dependent factors included vital sign values (max, mean, median, min), demographics (operation history, previous admission, age, sex), and the presence of symptoms.

## Results

Among the included patients, 4,660 patients were identified and 3,996 patients remained following exclusion criteria regarding age and data integrity. The baseline characteristics were classified by the maximum COVID-19 severity during admission. (Table 1) Older age, presence of underlying disease, previous operation, and previous admission were associated with severe COVID-19 outcomes. The median duration of hospital admission was 9 days (interquartile range: 7–12 days; S4 Fig). A good example of the model prediction is provided in Fig 1. The model can predict the aggravation and recovery of COVID-19 1 and 2 days in advance.

### Model performances

The model predicted the disease severity per day, including the present day (day 0), day 1, and day 2. The random forest model had the best performance, followed by the XG boost model,

**Table 1. Baseline characteristics of included patients.**

| Severity | Mild | Moderate | Severe | p-value |
|---|---|---|---|---|
| | (N = 2704) | (N = 955) | (N = 337) | |
| Age (years) | 49.9 ± 17.0 | 58.4 ± 16.3 | 66.5 ± 15.0 | <0.005 |
| Sex (male) | 1202 (44.5%) | 499 (52.3%) | 214 (63.5%) | <0.005 |
| Smoking | 346 (12.8%) | 73 (7.6%) | 38 (11.3%) | <0.005 |
| Alcohol | 736 (27.2%) | 208 (21.8%) | 70 (20.8%) | 0.001 |
| Weight (kg) | 66.1 ± 15.1 | 69.2 ± 16.7 | 67.9 ± 15.4 | <0.005 |
| Height (cm) | 164.8 ± 10.8 | 164.3 ± 11.4 | 164.2 ± 9.1 | 0.374 |
| BMI (kg/m$^2$) | 26.1 ± 61.9 | 26.4 ± 19.4 | 25.0 ± 4.3 | 0.906 |
| Previous medical history | | | | |
| Hypertension | 609 (22.5%) | 343 (35.9%) | 164 (48.7%) | <0.005 |
| Diabetes mellitus | 294 (10.9%) | 213 (22.3%) | 99 (29.4%) | <0.005 |
| Tuberculosis | 37 (1.4%) | 16 (1.7%) | 8 (2.4%) | 0.333 |
| Dyslipidemia | 208 (7.7%) | 97 (10.2%) | 35 (0.4%) | 0.028 |
| Congestive heart disease | 77 (2.8%) | 23 (2.4%) | 19 (5.6%) | 0.009 |
| Neurologic disease | 67 (2.5%) | 55 (5.8%) | 25 (7.4%) | <0.005 |
| Pulmonary disease | 56 (2.1%) | 19 (2.0%) | 11 (3.3%) | 0.336 |
| Liver disease | 32 (1.2%) | 17 (1.8%) | 6 (1.8%) | 0.318 |
| Other diseases | 1089 (40.3%) | 404 (42.3%) | 187 (55.5%) | <0.005 |
| Operation history | 875 (32.4%) | 319 (33.4%) | 135 (40.1%) | 0.018 |
| Previous admission | 716 (26.5%) | 271 (28.4%) | 115 (34.1%) | 0.01 |

The previous admission denotes all the previous admission in the BMC regardless of the period. The severity of the patients was classified at the patient-level and the highest severity during the admission was selected for the classification.

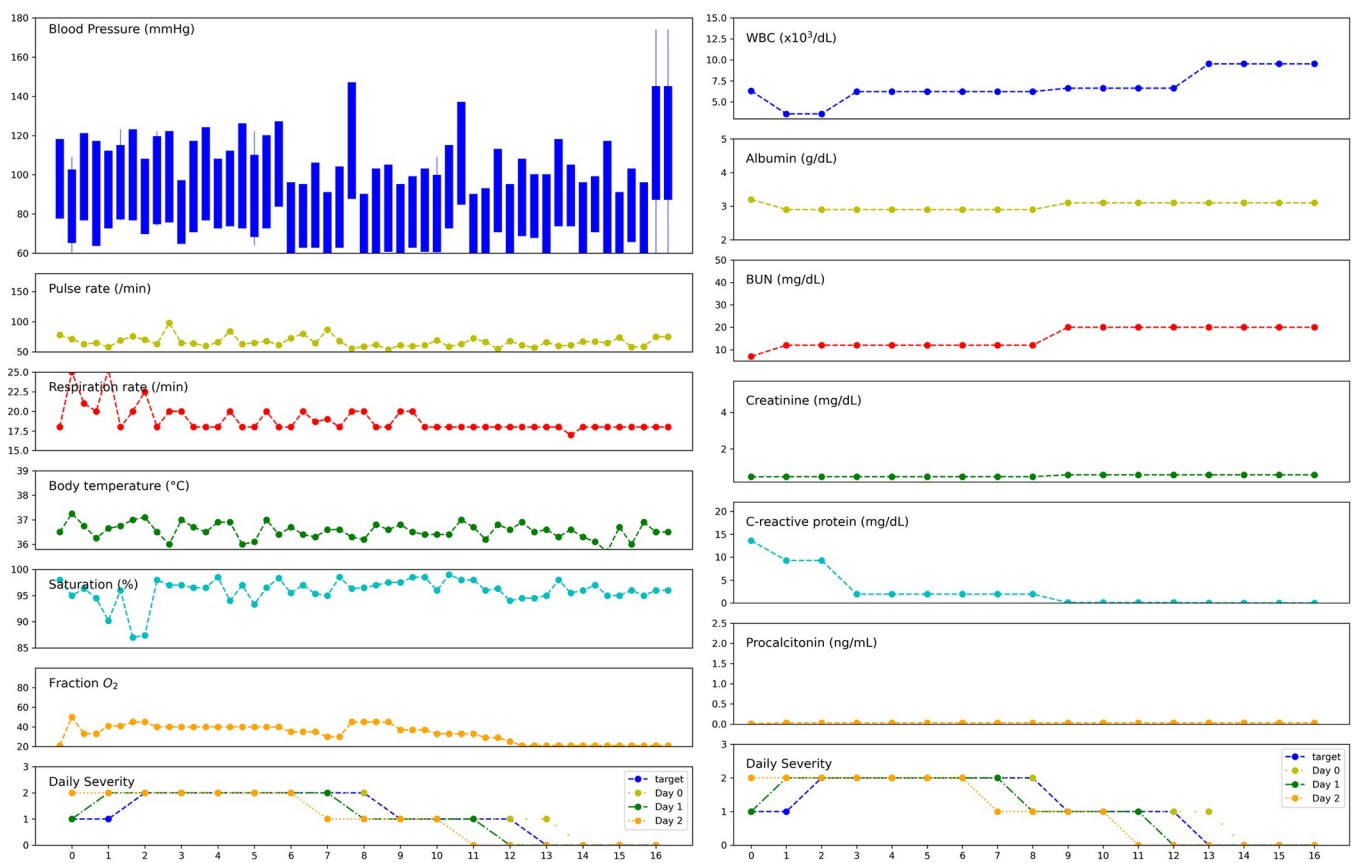

**Fig 1. Example of prediction of daily severity during admission.** The target is actual daily severity, and day 0 to day 2 represent the model predictions. The patient's status aggravated to severe on day 2, recovered to moderate on day 9, and became mild on day 13. The model (day 2) predicts disease aggravation and recovery 2 days before the infection.

the deep neural network model, and the logistic regression model. The model's prediction performance peaked at day 0 with an AUROC of 0.981 and an AUPRC of 0.940 (DNN), and it decreased as the prediction horizon expanded (Table 2, S3 Table).

**Table 2. The best model performance for daily COVID-19 severity.**

|  | Models | Accuracy (%) | AUROC | AUPRC |
|---|---|---|---|---|
| Day 0 | Logistic regression | 89 [88.4–89.6] | 0.962 [0.959–0.966] | 0.872 [0.861–0.883] |
|  | DNN | 92.9 [92.3–93.4] | 0.972 [0.969–0.976] | 0.934 [0.926–0.943] |
|  | XG boost | 95.7 [95.3–96.1] | 0.991 [0.99–0.993] | 0.972 [0.967–0.977] |
|  | Random forest | 95.2 [0.966–0.861] | 0.992 [0.99–0.993] | 0.977 [0.973–0.98] |
| Day 1 | Logistic regression | 87.1 [86.4–87.7] | 0.948 [0.944–0.952] | 0.837 [0.826–0.849] |
|  | DNN | 91.9 [91.3–92.4] | 0.966 [0.962–0.97] | 0.923 [0.913–0.931] |
|  | XG boost | 94.4 [94–94.9] | 0.985 [0.982–94.9] | 0.958 [0.951–0.963] |
|  | Random forest | 94.4 [93.9–94.9] | 0.988 [0.987–0.99] | 0.969 [0.965–0.973] |
| Day 2 | Logistic regression | 86.2 [85.5–86.9] | 0.938 [0.934–0.942] | 0.808 [0.795–0.82] |
|  | DNN | 91.6 [91.1–92.2] | 0.962 [0.958–0.966] | 0.913 [0.904–0.921] |
|  | XG boost | 93.6 [93.1–94.1] | 0.979 [0.976–0.982] | 0.946 [0.939–0.953] |
|  | Random forest | 93.9 [93.4–94.4] | 0.985 [0.983–0.987] | 0.959 [0.953–0.964] |

**Table 3. Comparison of the model performances between DNN and transformer.**

| Prediction horizon (days) | Models | Input lengths (days) | Accuracy | AUROC | AUPRC |
|---|---|---|---|---|---|
| 0 | DNN | 1 | 93.45 | 0.981 | 0.94 |
| | Transformer | 2 | 91.33 | 0.981 | 0.938 |
| | | 6 | 93.44 | 0.984 | 0.945 |
| | | 10 | 93.12 | 0.983 | 0.94 |
| | | 14 | 92.51 | 0.982 | 0.936 |
| | | 20 | 93.22 | 0.984 | 0.943 |
| 1 | DNN | 1 | 92.62 | 0.969 | 0.914 |
| | Transformer | 2 | 89.49 | 0.967 | 0.909 |
| | | 6 | 89.72 | 0.967 | 0.904 |
| | | 10 | 90.17 | 0.968 | 0.91 |
| | | 14 | 88.29 | 0.963 | 0.891 |
| | | 20 | 88.46 | 0.963 | 0.897 |
| 2 | DNN | 1 | 91.82 | 0.963 | 0.907 |
| | Transformer | 2 | 85.08 | 0.952 | 0.875 |
| | | 6 | 88.84 | 0.96 | 0.894 |
| | | 10 | 87.83 | 0.954 | 0.874 |
| | | 14 | 86.59 | 0.954 | 0.876 |
| | | 20 | 85.46 | 0.95 | 0.866 |

The accuracy of prediction decreased with increasing time horizon (accuracy: 92.6% [day 1], 91.8% [day 2], DNN). The transformer, which is more complex than DNN, did not show better performance. (Table 3) Regarding severity, the DNN model showed AUROC values of 0.958, 0.928, and 0.987 for mild, moderate, and severe COVID-19, respectively. In AUPRC, the model showed prediction performance of 0.985, 0.823, and 0.947 for mild, moderate, and severe COVID-19 after 2 days (Fig 2).

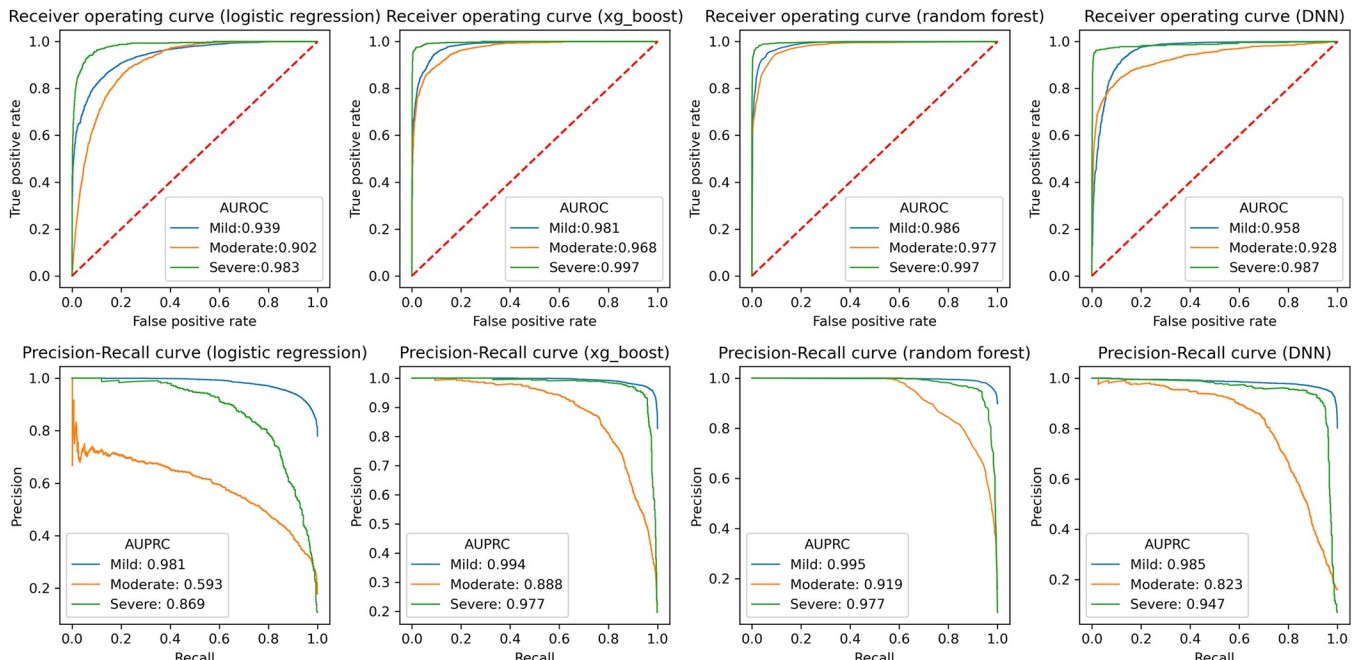

**Fig 2. The AUROC and AUPRC for daily prediction of COVID-19 severity.** The upper figures denote the receiver operating curve for predicting day 0, day 1, and day 2. The lower figures denote the precision-recall curve for each severity outcome.

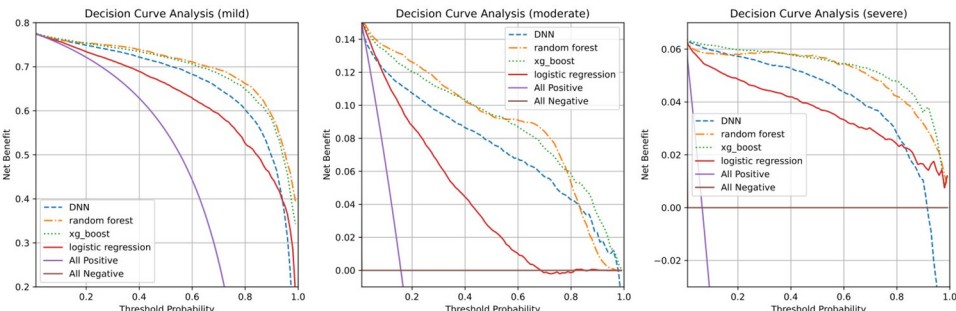

**Fig 3. The comparison of the model by decision curve analysis.** The figure represents the decision curve analysis for severity predictions for the period two days ahead. The baseline comparison (All positive) is premised on the assumption that every two-day event will be mild, moderate, or severe. All of our models demonstrated benefits exceeding the basic assumption made without any model (All positive).

In addition, our decision curve analysis highlighted the net-benefit of the model based on varying threshold probabilities. Consistent with our previous AUROC comparisons, the random forest and XG boost models demonstrated higher net-benefits for the mild, moderate, and severe predictions for day 2, as displayed in Fig 3.

## Feature importance of different models

The SHAP method was used to demonstrate the models' feature importance. (Fig 4) We selected the highest feature importance up to the 20th variable for predicting severe COVID-19 after 2 days. Although feature importance was determined in the same subset of the test set, the selected features varied among models. DNN showed higher feature importance in the following order: operation history, male sex, older age, hypertension, and nausea. The XG boost model used the following ranking: steroid (maximum, mean), lactate dehydrogenase (minimum), old age, and SpO2 (minimum, count). Additionally, the random forest model used the following: steroid (maximum), SpO2 (count), Pulse rate (count), steroid (minimum), and SpO2 (minimum).

The factors were categorized as patient-dependent and hospital/treatment-dependent. For predicting patient's severity, the patient factor cannot be altered by the hospital where the

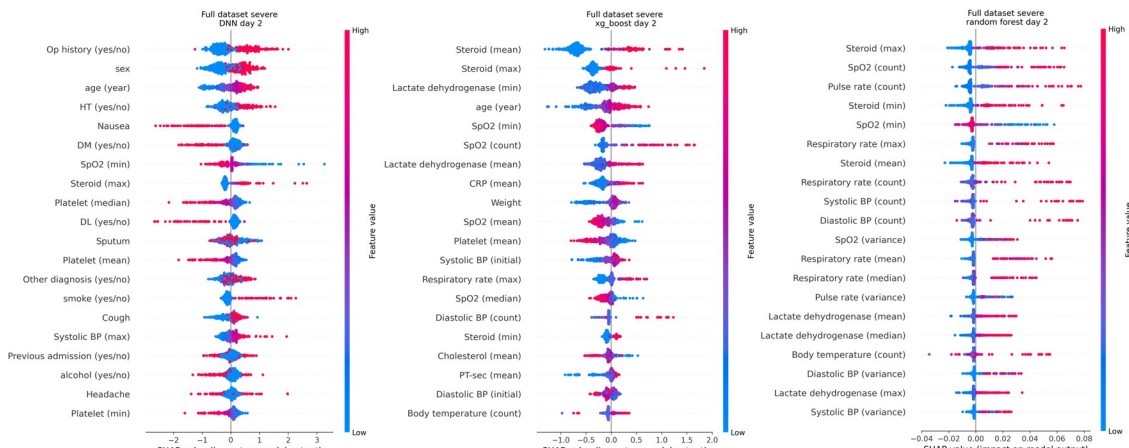

**Fig 4. The feature importance among the models.** The three model's feature importance were compared by the SHAP method. The tree-based model places a greater focus on steroid usage, SpO2 count, and vital signs (Pulse rates, and diastolic BP). The DNN model placed a greater focus on demographic data and symptoms.

patient was admitted. However, hospital/treatment factors, such as the number of vital signs that should be monitored and the dosage of steroid administered, can differ between hospitals. Regarding the SHAP variable analysis, the patient-dependent percentage for each model was 95% in DNN, 75% in XGBoost, and 35% in random forest (p-value: 0.065). Thus, the DNN prioritized patient factors, whereas the random forest and XG boost prioritized hospital/treatment factors. The same phenomenon was also observed for predicting mild and moderate COVID-19 infection (S5 Fig).

## Discussion

We examined a range of models to forecast daily COVID-19 severity. The DNN model had an accuracy, AUROC, and AUPRC of 91.8%, 0.96, and 0.90, respectively for predicting the severity of COVID-19 as either mild, moderate, or severe 2 days in advance. The transformer model that received trend data did not perform better than the DNN model that received only daily data. The tree model (random forest, XG boost) outperformed DNN; however, the model's feature importance was heavily weighted towards hospital/treatment factors, resulting in unclear generalizability in other hospitals.

Severe COVID-19 shows a similar disease course as that observed in acute respiratory distress syndrome, which requires high-level oxygen support from high-flow nasal oxygen, ventilator support, and ICU care [25]. Respiratory devices are restricted resources; thus, treatment of severe COVID-19 patients is limited by device and healthcare worker availability [6]. Our model can help identify patients that show aggravation of the infection and recovery in near future; thus, clinicians could be advised to start or wean respiratory devices in a limited resource setting. Previous studies focused on predicting disease aggravation at admission along with real-time prediction [16, 26]. While it is essential to identify potential patients with infection aggravation in future, identifying those who will recover is also important in managing overall hospital resources. By integrating our model into clinical practice, clinicians can readily monitor patients who may be at risk of deterioration due to COVID-19 as well as those showing signs of recovery. Such insights prove valuable in anticipating potential strains on medical facilities and allow for proactive arrangements, including network preparations for transferring patients requiring ventilator support and intensive care unit (ICU) facilities.

A static prediction implies prediction of any future event at the onset of specific time [15]. For example, a logistic model incorporating clinical symptoms and abnormal laboratory tests at admission could predict whether patients would develop a serious illness within 30 days [10]. Dynamic or real-time prediction predicts outcomes at each time-point [15]. Recently, a dynamic prediction model for forecasting future COVID-19-related aggravation [16] or mortality [11] was developed. Although temporal model (RNN) improved the performance of numerous tasks [13, 27, 28], daily severity prediction tasks did not benefit from the use of temporal model in this study. We anticipated improved performance by incorporating temporal information into the transformer model, but the improvement was task-dependent. In this study, the AUROC and AUPRC for predicting severity 2 days in advance were >0.9, which could indicate that the task was rather simple and that the trend information provides no additional information compared to daily information.

Among non-temporal models, tree-based models, such as random forest and XG boost, performed better than DNN and logistic regression models. When logistic regression demonstrates performance equivalent to that of machine learning models, including deep learning model, a basic model may be preferred over a machine learning model because it is more interpretable and understandable [29]. In this study, as the dimension of the input data exceeded 300, machine learning enhanced model performance, suggesting that complex computation is

more advantageous for prediction tasks than linear computation [30]. Regarding model performance, the random forest model should be preferred because it exhibited the best results among non-temporal models. However, the SHAP method evaluated model reliability based on the significance of features rather than model performance. The tree model revealed that drug use and frequency of vital sign checks are of higher importance, and these factors are dependent on treatment pattern and hospital rules. Moreover, in random forest and XG boost models, steroid usage was the most important factor. Steroid usage was considered a surrogate marker for severe COVID-19 infection, even if it is intended to treat the disease. Therefore, performance may be diminished if the treatment pattern or hospital rule for vital sign changes. This risk was identified in a prior study that employed the ensemble model based on two tree models; higher internal validation performance was not maintained in other hospitals [31]. This is because the ensemble model was hospital- and treatment-dependent, and the prediction was not transferable to other hospitals. However, in this study, the deep learning model placed greater emphasis on patient demographics and symptoms. As the feature cannot be altered even if the patient is transferred to a different hospital, the model's predictions are transferable to other hospital settings. Thus, we determined that the DNN model is superior than the tree model due to greater generalizability despite its inferior performance.

The strength of this study is that our deep learning model predicts COVID-19 severity regardless of the phase of the disease. To the best of our knowledge, this is the first study to predict future severity using a time-series dataset, regardless of the course of COVID-19. We assessed a variety of models, including machine learning and deep learning models, as well as data structures derived from non-temporal and temporal input. The model's interpretation was evaluated using the SHAP method, which considers each model's important features and their generalizability using a domain-wise knowledge approach. Furthermore, our model's predictive capability is particularly beneficial during nighttime when timely interventions are pivotal. By accurately estimating the potential risk of patient deterioration, clinicians are better equipped to allocate essential medical devices, ensuring that patients receive the necessary support even during off-peak hours

This study had some limitations. First, it was based on single-centered retrospective cohort data; thus, the model may not show high performance using other hospital datasets. Due to the difficulty of accessing other hospitals with a comparable patient group, external validation was restricted. The variability in the frequency of vital sign and laboratory tests may result in a higher incidence of missing data. Additionally, the unavailability of daily symptom records in other hospitals could potentially present challenges to the generalizability of our model. Nevertheless, we evaluated the model's essential components using the SHAP method and its generalizability with variable characteristics. We followed to a common technique for processing time-series data; thus, the problem regarding hospital and treatment-related factors would be similar to that of another task prediction. Consequently, our method for determining the importance of a variable will benefit from a future study focusing on generalizability. Further validation is required to support our findings. Second, the severity of COVID-19 was determined by WHO guidelines and the application of oxygen. The application of oxygen could depend on the clinician's preference. If the clinical pattern of oxygen usage in another hospital differed from that in ours, our model would not accurately predict the application and weaning of oxygen at that hospital. If severity could be independently established as per the physician's preference, the target severity would be appropriate for any future research. Third, this study has not yet undergone prospective validation, hence, it does not provide guidance on what steps should be taken in instances where there is a discrepancy between clinical judgments and model predictions. A subsequent prospective study is needed to collect actual clinical experiences and reassess how to proceed in such scenarios.

## Conclusions

Our hierarchical transformer model showed higher predictive performance in the early and late periods of hospital admission. Our model could be useful for COVID-19 patients by predicting their future outcomes and aids the distribution of respiratory devices.

## Supporting information

**S1 File. Supplement methods.**
(DOCX)

**S1 Fig. Missing value proportion per patient.**
(TIF)

**S2 Fig. Exploration of laboratory distribution.**
(TIF)

**S3 Fig. Difference of mode, median, and mean in ferritin value.**
(TIF)

**S4 Fig. Histogram of the length of hospital admission duration.**
(TIF)

**S5 Fig. The feature importance of the models for predicting mild and moderate COVID-19.**
(TIF)

**S1 Table. Percentile and clipping values of continuous variables.**
(DOCX)

**S2 Table. Symptom presentation of COVID-19 patients during hospital admission.**
(DOCX)

**S3 Table. Detailed model performance comparison.**
(DOCX)

## Author Contributions

**Conceptualization:** Hyungjun Park, Chang-Min Choi, Sung-Hoon Kim, Su Hwan Kim, Deog Kyoem Kim, Ji Bong Jeong.

**Data curation:** Su Hwan Kim, Deog Kyoem Kim, Ji Bong Jeong.

**Formal analysis:** Hyungjun Park.

**Funding acquisition:** Sung-Hoon Kim.

**Investigation:** Hyungjun Park.

**Methodology:** Hyungjun Park.

**Software:** Hyungjun Park.

**Supervision:** Sung-Hoon Kim, Su Hwan Kim, Deog Kyoem Kim.

**Validation:** Hyungjun Park, Chang-Min Choi.

**Visualization:** Hyungjun Park.

**Writing – original draft:** Hyungjun Park.

**Writing – review & editing:** Hyungjun Park, Ji Bong Jeong.

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
