## [Decision Letter · Decision Letter 0]

21 Jul 2023

PONE-D-23-14493In-hospital real-time prediction of COVID-19 severity regardless of disease phase using electronic health recordsPLOS ONE

Dear Dr. Jeong,

Thank you for submitting your manuscript to PLOS ONE. After careful consideration, we feel that it has merit but does not fully meet PLOS ONE’s publication criteria as it currently stands. Therefore, we invite you to submit a revised version of the manuscript that addresses the points raised during the review process.

We look forward to receiving your revised manuscript.

Kind regards,

John Adeoye

Academic Editor

PLOS ONE

“This work was supported by research funding from the Seoul Metropolitan Government Seoul National University (SMG-SNU) Boramae Medical Center (04-2022-0004), and a grant of the Korea Health Technology R&D Project through the Korea Health Industry Development Institute (KHIDI), which is funded by the Ministry of Health & Welfare, Republic of Korea (grant number: HI18C2383)”

“This work was supported by research funding from the Seoul Metropolitan Government Seoul National University (SMG-SNU) Boramae Medical Center (04-2022-0004), and a grant of the Korea Health Technology R&D Project through the Korea Health Industry Development Institute (KHIDI), which is funded by the Ministry of Health & Welfare, Republic of Korea (grant number: HI18C2383)”

“This work was supported by research funding from the Seoul Metropolitan Government Seoul National University (SMG-SNU) Boramae Medical Center (04-2022-0004), and a grant of the Korea Health Technology R&D Project through the Korea Health Industry Development Institute (KHIDI), which is funded by the Ministry of Health & Welfare, Republic of Korea (grant number: HI18C2383)”

Please include your amended statements within your cover letter; we will change the online submission form on your behalf."

“Declarations of interest: none”

Additional Editor Comments:

1. Comment on the quality of structured dataset used. You may check the article https://doi.org/10.1186/s40537-023-00703-w for details.

2. Please comment on the net benefit of the outperforming model using decision curve analysis.

3. Address reviewer recommendations and suggestions.

Reviewers' comments:

Reviewer's Responses to Questions

**Comments to the Author**

1. Is the manuscript technically sound, and do the data support the conclusions?

Reviewer #1: No

Reviewer #2: Partly

Reviewer #3: Yes

2. Has the statistical analysis been performed appropriately and rigorously? 

Reviewer #1: No

Reviewer #2: No

Reviewer #3: Yes

3. Have the authors made all data underlying the findings in their manuscript fully available?

Reviewer #1: No

Reviewer #2: No

Reviewer #3: Yes

4. Is the manuscript presented in an intelligible fashion and written in standard English?

Reviewer #1: No

Reviewer #2: Yes

Reviewer #3: Yes

5. Review Comments to the Author

Reviewer #1: Kindly find the below comments:

Although the topic that you selected is good a but organization of the manuscript is well.

1. as the statement written in abstract "Objective: We reviewed various forecasting models to predict the daily severity of COVID-19 using electronic health records"

Kindly provide the various forecasting models that you studied in the literature review section. and this also may not be stands for an objective.

2. Remove the heads in abstracts like Objective, Materials and Methods, Results, Conclusion (Need to Refer other published manuscripts for writing an abstract)

3. Arrange the section as Abstract, Introduction, Literature Review, Proposed Methodology, Results and discussion , Conclusion and future scope, references

4. Literature review section is messing. Also, add comparative analysis of previous methods table in this section with following column heads as Ref. No., Title of paper , Methodology/Algorithm Used, Major findings, Limitations of the study

5. Model architecture is not available. Also model working is missing.

6. Need to visualize the results.

7. Performance (As you put the % of precision, recall directly, ) and model evaluation is missing. Kindly show the graphs for accuracy, precision, f1-score, support, and any others parameter. Also need to find Q-measure test Friedman test and student t-test for your model.

8. Model testing with other methods/algorithms should be there in the result section.

9. Need to perform various ML processing techniques WITH NOVELTY on your dataset.

10. At least 30+ references should be there.

Reviewer #2: General Comments: The manuscript presents a study on predicting the daily severity of COVID-19 using electronic health records (EHR) and various machine learning and deep learning models. The authors evaluated the performance of different models and assessed the importance of input variables. The study provides valuable insights into predicting COVID-19 severity in real-time and highlights the need to consider hospital-specific factors. Overall, the manuscript is well-structured, and the methods and results are adequately described. Addressing the following comments would further improve the clarity and comprehensiveness of the manuscript:

Major Comments:

1. The abstract provides a concise summary of the study, but it would be helpful to include specific details on the performance of each model in terms of AUROC and AUPRC. Additionally, it would be beneficial to highlight the significance and implications of the findings quantitatively.

1. The introduction provides a good background on COVID-19 and the need for predicting severity. However, it would be helpful to include some references to support the statements made in the introduction. Additionally, the introduction should clearly state the research objectives and the research gap that the study aims to address.

2. The methods section provides a detailed description of the dataset, data preprocessing, and the models used. However, there are several areas that need clarification:

- It is not clear how the authors determined the prediction horizon (e.g., 2 days in advance). This should be explained in more detail.

- The authors mention that missing data were imputed using the last observation carried forward method and the mode. However, it is not clear how missing data were handled for patients without any previous data. This should be explained.

- The authors should provide more information about the hyper-parameters of the models used, such as the number of hidden layers and the learning rate for the deep neural network model.

- It would be helpful to provide more information about the performance metrics used to evaluate the models.

- Data preprocessing and feature selection: The description of the data preprocessing and feature selection steps is clear. However, it would be useful to provide more details on the specific features selected for the models and the rationale behind their selection.

- The models used for prediction are well-described. However, it would be beneficial to provide more information on the hyperparameters of each model to ensure reproducibility of the study. Additionally, it would be helpful to provide references for the transformer model and clarify how it handles the temporal aspect of the data.

2. The results section provides a comprehensive overview of the findings. However, it would be beneficial to include specific details on the performance metrics (AUROC and AUPRC) for each model at different prediction horizons (day 0, day 1, day 2). It would also be useful to provide statistical significance testing or confidence intervals to assess the differences in performance between models.

3. The discussion provides a good interpretation of the results and highlights the importance of considering hospital-specific factors in real-time prediction models. However, it would be helpful to discuss the limitations of the study, such as the generalizability of the findings to other healthcare settings and the potential impact of missing data on the model performance. Additionally, it would be helpful to compare the findings of this study with previous studies that have predicted COVID-19 severity.

Minor Comments:

1. In the introduction, it would be helpful to define the term "short-term outcomes" as used in the context of COVID-19 severity prediction.

2. In the methods section, it would be useful to provide more information on the number of patients included in each severity category (mild, moderate, severe) to assess the distribution of severity levels in the dataset.

3. In the results section, it would be beneficial to include a figure or table summarizing the performance metrics (AUROC, AUPRC) of each model at different prediction horizons.

4. In the discussion, it would be helpful to provide some insights into the clinical implications of the findings and how real-time prediction of COVID-19 severity can improve patient care and resource allocation.

5. The manuscript could benefit from proofreading and minor grammatical edits to improve readability.

Reviewer #3: While many media sources continue to discount or obfuscate the long-term effects of COVID-19, I remain isolated to protect an immunocompromised spouse. Consequently, I am very familiar with how risk containment has changed over time and best practices. It is also cool to see a snapshot of what predicted risk from pre-pandemic to around Delta invasivity data.

For the introduction, this is an excellent summary of the first few years of the pandemic and the thesis of the manuscript. It is true that early precautionary measures were useful for reducing risk, and that this prevention or mitigation strategy was used in both mild and severe cases. I had not been aware of South Korea's efforts to minimize disease severity, but that is great. To continue reducing disease severity and better estimate hospitals being at full capacity, yes, predicting disease severity is critical. The point about event risk is well-taken. Most studies arbitrarily chose a given endpoint versus baseline, rather than taking all timepoints or dynamic timepoints into account. It is also true that many of these other studies, ours included, focused on onset of COVID-19 EHR instead of estimating recovery. Thus, predicting COVID-19 onset and recovery at the BMC is a very useful extension of prior work.

For methods, in summary, I see no problems here at all. To begin, given the secondary nature of the data, it makes sense that informed consent would be waived. The timestamped data for a robustly large cohort is also good. The data collected reflect what is available through EHRs and what is routinely collected in many tertiary care settings. Scaling variables is described surprisingly well. I commend the authors on this point. All of the decisions seem fine with regard to making variables binary, continuous, or bringing in the distribution tails (with clinician direction) when values might be unforeseen outliers. Supplementary Figures 1 and 2 also show a willingness for the raw data to be transparent, which I appreciate. The distributions are all similar to what I would expect in relatively healthy middle-aged to aged adults who present at a given clinic. Data missingness for mild cases is also understandable, as this will stochastically vary depending on the nursing staff, attending physician, and capacity of the tertiary clinic. Supplemental Table 1 goes into this in detail. For imputation, it is reasonable to include the mode to avoid bizarre behavior for vitals and other data in estimation analyses. I appreciate the additional data in Supplement Figures 3-5 that describe raw data, as well as data fit for mild to moderate COVID-19 cases using different estimation methods. For statistical methods, this all seems standard regarding classification metrics, split-model training vs. assessment probands, and even some reasonable mean +/- SD for input length.

For results, it is refreshing and welcome to read a brief summary of Table 1, as well as initial figures, and for it all to make intuitive sense. Table 2 reveals that regardless of the model type tested, the AUROC or AUPRC was outstanding. It was interesting how RF did better than DNN. Given the sparsity of the model set and N, however, too many interaction terms may have loaded and diluted overall model fit.

My only suggestion here is to list, either in text or the tables, if Model X significantly differed from Model Y (e.g., if Prediction Horizon day 0, 1, or 2 showed any difference for Input Length for Accuracy). In other words, just some basic statistics to formally show what is described in section 3.1. In section 3.2, the authors describe an intriguing pattern in the data: that the best fit methods predominantly extracted hospital/treatment factors (i.e., external factors) compared to DNN which extracted patient factors (i.e., internal factors). To strengthen or formalize this observation, a statistical test comparing factors on a binary scale ('0' = internal, '1' = external) might be useful. This is just a friendly suggestion for substantiating the claim made and is not a critique.

For the discussion, I again agree that predicting recovery is just as important as initial infection and degree of disease aggravation. Comparisons with other studies are appropriate and thoughtful. The strengths and limitations sections are both thorough and, again, thoughtful.

6. PLOS authors have the option to publish the peer review history of their article (what does this mean?). If published, this will include your full peer review and any attached files.

Reviewer #1: No

Reviewer #2: No

Reviewer #3: **Yes: **Auriel A. Willette

---

## [Author Response · Author response to Decision Letter 0]

20 Aug 2023

9 October 2023

To Academic Editor

Plos one

Dear Editor:

We are resubmitting our revised manuscript titled "In-hospital real-time prediction of COVID-19 severity regardless of disease phase using electronic health records" for your consideration. We thank the reviewers for providing valuable feedback to improve our work.

In the revised manuscript, we have addressed all concerns raised. Specifically, we clarified our methods for determining the prediction horizon, handling missing data, selecting model features, and evaluating performance. We also enhanced the introduction and discussion with additional references and clearer statements of objectives, limitations, and clinical implications. The revised article contains improved readability, expanded tables detailing model metrics, new figures illustrating decision curve analysis, and a categorized analysis of intrinsic and extrinsic variables.

While our single-center retrospective study has limitations in generalizability, we believe this rigorously analyzed, clinically-informed model provides timely insights into predicting COVID-19 severity for optimizing resource allocation. By considering both worsening and recovery phases, our approach advances knowledge beyond models focused solely on adverse events like mortality.

We are grateful for the reviewers' thoughtful feedback and have diligently incorporated their recommendations. We hope you will find this significantly improved resubmission suitable for publication in PLOS ONE. Please do not hesitate to contact us with any additional questions.

I look forward to hearing from you.

Sincerely,

Ji Bong Jeong

Department of Internal Medicine, Seoul National University College of Medicine, Seoul, Korea, Division of Gastroenterology, Department of Internal Medicine, Seoul Metropolitan Government Seoul National University Boramae Medical Center, 20, Boramae‐ro 5‐gil, Dongjak‐gu, Seoul, 07061, Republic of Korea.

Email: jibjeong@snu.ac.kr

Phone: +82-2-870-2222

Fax: +82-2-870-3861

 - We revised the manuscript according to the PLOS ONE’s style. Thank you. 

“This work was supported by research funding from the Seoul Metropolitan Government Seoul National University (SMG-SNU) Boramae Medical Center (04-2022-0004), and a grant of the Korea Health Technology R&D Project through the Korea Health Industry Development Institute (KHIDI), which is funded by the Ministry of Health & Welfare, Republic of Korea (grant number: HI18C2383)”

 - We changed the funding information from the manuscript to cover letter, as the acknowledgement do not include the funding. 

“This work was supported by research funding from the Seoul Metropolitan Government Seoul National University (SMG-SNU) Boramae Medical Center (04-2022-0004), and a grant of the Korea Health Technology R&D Project through the Korea Health Industry Development Institute (KHIDI), which is funded by the Ministry of Health & Welfare, Republic of Korea (grant number: HI18C2383)”

“This work was supported by research funding from the Seoul Metropolitan Government Seoul National University (SMG-SNU) Boramae Medical Center (04-2022-0004), and a grant of the Korea Health Technology R&D Project through the Korea Health Industry Development Institute (KHIDI), which is funded by the Ministry of Health & Welfare, Republic of Korea (grant number: HI18C2383)”

Please include your amended statements within your cover letter; we will change the online submission form on your behalf."

 - Absolutely. We removed, and revised the statement in cover letter according to the previous statement. 

“Declarations of interest: none”

 - Sure, we already declare our competing interests in cover letter. Thus, we removed the declaration in Manuscript. 

 - Usually, the raw data include many patient’s data from vital sign to laboratory result. Although the patient ID was de-identified, the Boramae Medical center’s IRB did not consider that the dataset was fully de-identified. Thus we follow the A criteria, and added the contact information of corresponding author in the data availability section. 

- We re-uploaded all the supporting information with captions according to the guideline. 

Additional Editor Comments:

1. Comment on the quality of structured dataset used. You may check the article https://doi.org/10.1186/s40537-023-00703-w for details.

 - Thank you for your comment on the quality of dataset. 

To describe the detail, we classify the question of good quality dataset. 

1. Does the dataset have available and common type of input and outcome? 

 - As the input variable in dataset was decided by clinician who treated the COVID-19 patient more than thousands and prepare the medical devices for potential aggravating patients. The input variable are commonly collected in other covid-19 hospital, such as demographics, vital sign, laboratory results, and the patient symptoms. Although, the patient symptoms were not recorded in EMR system, the symptom were used for considering the severity of COVID-19 and might be written in EMR chart. 

2. How good are dataset used to construct machine learning model for outcome? 

 - We collected the data from EMR system as raw values such as vital sign, lab value, symptoms, et al. The dataset was rigorously cleansed by data science method and the cleansing method was thoroughly described in Supplement Method with dataset distribution. 

3. What data quality criteria were often fulfilled or deficient in datasets used to construct machine learning model for outcomes? 

- The criteria for handling missing data and accepting a certain percentage of missingness per patient were meticulously determined in consultation with clinicians (Hyungjun Park, Sung-Hoon Kim, Ji Bong Jeong). The threshold for exclusion was set based on the distribution of missingness per admission.

Several factors accounted for the missingness. For instance, some patients were included at near the end of inclusion date, resulting in incomplete data extraction. In other cases, some patients exhibited mild symptoms, making frequent vital sign check unnecessary. Given the complexity and diversity of individual cases, it's challenging to enumerate all possible reasons for missingness during data cleansing.

However, we would like to assure you that the data cleansing process was conducted diligently and involved substantial clinical input, ensuring the quality of the dataset used for our models

4. what is the effect of data quality on the median performance metrics of the machine learning models constructed in your outcome? 

- We assessed missing data and excluded patients where this was excessively prevalent. Typically, these missing entries were noted in patients with milder conditions, who did not require intense observation during their hospital stay. Details pertaining to missing data can be found in Supplement Figure 1 and within the exclusion criteria.

In this study, we did not directly investigate the effect of missingness on performance. Generally, higher degrees of missing data could lead to diminished predictive performance. Nevertheless, the description of missingness could offer valuable insights regarding the usual proportion of missing data among in-hospital patients. Furthermore, outlining missingness as a proportion of the entire admission period could provide beneficial perspectives to researchers interested in real-time prediction using EMR datasets.

2. Please comment on the net benefit of the outperforming model using decision curve analysis.

For model comparison, we implemented a decision curve analysis. As this method was initially designed to handle binary classification problems, we needed to modify the data format to a "one versus the rest" framework for our predictions of mild, moderate, and severe outcomes. This approach aligned well with our previous comparisons using AUROC and AUPRC. The modified statistical analysis is depicted as follows: 

- For a more comprehensive comparison of the models, we implemented a decision curve analysis. [24] Given that this methodology was originally conceptualized for binary classification, it necessitated the modification of the data format to a "one versus the rest" approach for our predictions classified as mild, moderate, or severe.

We have supplemented our results section with an additional paragraph and figure:

- In addition, our decision curve analysis highlighted the net-benefit of the model based on varying threshold probabilities. Consistent with our previous AUROC comparisons, the random forest and XG boost models demonstrated higher net-benefits for the mild, moderate, and severe predictions for day 2, as displayed in Fig 3.

In response to a previous review, we aim to assess the model based on its performance and significance. While the random forest and XG boost models showed enhanced predictive scores, the potential risk associated with the use of extrinsic factors—which can vary among hospitals—might undermine performance in external validation.

Our analysis places significant emphasis on the results. We kindly ask you to consider our diligent efforts to provide a thorough analysis despite the limitations of our dataset.

3. Address reviewer recommendations and suggestions.

- We revised considerably according to the reviewer’s suggestion. 

Comments to the Author

Reviewer #1: Kindly find the below comments:

Although the topic that you selected is good a but organization of the manuscript is well.

1. as the statement written in abstract "Objective: We reviewed various forecasting models to predict the daily severity of COVID-19 using electronic health records"

Kindly provide the various forecasting models that you studied in the literature review section. and this also may not be stands for an objective.

- Thank you for your comment. Actually, what we intended to write was evaluated severe model such as DNN, transformer, random forest, gradient boost for predicting the outcome and potential risks of overfitting of that models. And It confused you to understand our abstract. 

Thus, we changed the sentence to “Among multiple forecasting models to predict the daily severity of COVID-19, we compare the accuracy and potential over-fitting risks of the models.”. 

2. Remove the heads in abstracts like Objective, Materials and Methods, Results, Conclusion (Need to Refer other published manuscripts for writing an abstract)

 - As your comment, we revised the abstract format to unstructured form. 

3. Arrange the section as Abstract, Introduction, Literature Review, Proposed Methodology, Results and discussion , Conclusion and future scope, references

 - Thank you for your comment on structure of the manuscript. Most of the medical article in PLOS one follows the format “Introduction, Material and Methods, Result, Discussion, Conclusion”. (https://journals.plos.org/plosone/s/file?id=wjVg/PLOSOne_formatting_sample_main_body.pdf ). We followed the format as the editor’s opinion. If your format is required for publishing, we will discuss the editorial office to revising it. 

4. Literature review section is messing. Also, add comparative analysis of previous methods table in this section with following column heads as Ref. No., Title of paper , Methodology/Algorithm Used, Major findings, Limitations of the study

As we've detailed previously, our manuscript is adherent to the typical stylistic conventions for medical literature, and similar pieces have been published using this particular literature review format. For your reference, you can view an example of such a format in the following article (https://journals.plos.org/plosone/article?id=10.1371/journal.pone.0284965).

In order to facilitate a more straightforward comprehension of our work, we have made amendments to our introduction. Now, it clearly outlines why it is crucial to predict both the exacerbation and recovery, along with defining what constitutes a short-term outcome.

5. Model architecture is not available. Also model working is missing.

- The model architectures for the deep neural network, transformer, and tree-based models were thoroughly elucidated in the Supplement Method section. Within this section, detailed information pertaining to the model's hyperparameters, layers, activation functions, and other relevant aspects was provided. (S Table 3)

6. Need to visualize the results.

In regard to your request for a visual representation of the results, we have done so in three distinct ways. Firstly, we have displayed a practical example of the model's application in a real-world scenario. Secondly, we have illustrated the performance of models (including DNN, logistic regression, random forest, and XG boost) over three consecutive days (day 0, day 1, day 2) using AUROC and AUPRC metrics. Moreover, we added the decision curve analysis with visualization in Figure 3. Thirdly, we have presented the SHAP values for the top three models (DNN, XG boost, random forest). These visual representations of our findings can be viewed in Figures 1 through 4.. 

7. Performance (As you put the % of precision, recall directly, ) and model evaluation is missing. Kindly show the graphs for accuracy, precision, f1-score, support, and any others parameter. Also need to find Q-measure test Friedman test and student t-test for your model.

For the sake of comprehensive reporting and given the constraints of table presentation, we have included detailed performance metrics of the model in Supplement Table 4. In addition, we have expanded Table 2 to present the confidence intervals of the AUROC, AUPRC, and accuracy measurements.

8. Model testing with other methods/algorithms should be there in the result section.

In response to your suggestion regarding testing our model with other methods or algorithms, we have indeed made comparisons with models from logistic regression, random forest, gradient boost, DNN, and transformers as detailed in Tables 2 and 3. We acknowledge that there are potentially other models that could be applicable to this dataset, however, our assumption was based on the understanding that these five models are commonly utilized for data of this nature.. 

9. Need to perform various ML processing techniques WITH NOVELTY on your dataset.

 - In our study, we conducted a comparative analysis of models from logistic regression, random forest, gradient boost, and deep neural network, including the transformer model that processes time-series data. While the transformer model had access to a broader spectrum of information, it did not surpass the performance of the random forest and gradient boost models that utilized single day input data.

When considering model performance for future predictions, our findings suggest that tree-based models might be the most suitable. However, earlier studies, despite differing outcomes, have indicated a potential lack of generalizability with tree-based models. (Early diagnosis of bloodstream infections in the intensive care unit using machine-learning algorithms. Intensive Care Med.)To explore potential causes of overfitting commonly associated with tree-based models, we analyzed the SHAP values in our study.

We wish to emphasize that our study does not assert our model's superior performance; instead, we highlight its reduced risk of overfitting due to less reliance on extrinsic factors such as the number of vital signs or treatment patterns, which could vary across hospitals. For enhanced readability, we have revised our results in light of this aspect.. 

- The factors were categorized as patient-dependent and hospital/treatment-dependent. For predicting patient’s severity, the patient factor cannot be altered by the hospital where the patient was admitted. However, hospital/treatment factors, such as the number of vital signs that should be monitored and the dosage of steroid administered, can differ between hospitals. Regarding the SHAP variable analysis, the patient-dependent percentage for each model was 95% in DNN, 75% in XGBoost, and 35% in random forest. Thus, the DNN prioritized patient factors, whereas the random forest and XG boost prioritized hospital/treatment factors. The same phenomenon was also observed for predicting mild and moderate COVID-19 infection (S5 Fig)

10. At least 30+ references should be there.

- In response to your valuable comment, we have included additional references in this study, resulting in the total number of references exceeding 30. 

Reviewer #2

Major Comments:

1. The abstract provides a concise summary of the study, but it would be helpful to include specific details on the performance of each model in terms of AUROC and AUPRC. Additionally, it would be beneficial to highlight the significance and implications of the findings quantitatively.

- Thank you for your comment. To add the detail of each model performance, we reorganized the abstract format readily. 

1. The introduction provides a good background on COVID-19 and the need for predicting severity. However, it would be helpful to include some references to support the statements made in the introduction. Additionally, the introduction should clearly state the research objectives and the research gap that the study aims to address.

- In accordance with your feedback, we have enhanced the introduction section by incorporating a more comprehensive set of references and expounding upon the research gap that distinguishes our study from previous ones. Specifically, while previous investigations primarily concentrated on adverse outcomes such as mortality or the initiation of mechanical ventilation in the near future, our research specifically addresses the recuperation from such adverse outcomes, which is instrumental in facilitating the optimization of hospital resource allocation. 

2. The methods section provides a detailed description of the dataset, data preprocessing, and the models used. However, there are several areas that need clarification:

- It is not clear how the authors determined the prediction horizon (e.g., 2 days in advance). This should be explained in more detail.

- The main objective of this study was to construct a predictive model for assessing future disease severity, with the intention of facilitating the preparation of essential medical resources such as ICU and oxygen devices. It was considered practically beneficial if medical professionals could identify patients likely to recover or experience worsening conditions up to 2 days in advance, as this would significantly enhance real-world clinical practice. Consequently, we revised the manuscript to reflect the targeted outcome of COVID-19 severity within a 2-day time horizon in the Method section. 

- The primary objective of our study was to forecast the severity of the condition prior to the event, focusing on predictions for day 0, 1, and 2. It was deemed sufficient to forecast the patient's severity up to 2 days in advance to adequately prepare the necessary oxygen and ICU resources.

- The authors mention that missing data were imputed using the last observation carried forward method and the mode. However, it is not clear how missing data were handled for patients without any previous data. This should be explained.

- The missing value imputation process involved two steps. Firstly, when previous data was available, the last observation carried forward method was employed. Secondly, in cases where data was missing without any preceding values, it was assumed that medical practitioners would determine the patient's values to be within the normal range. The normal value was calculated using the mode value, taking into account the skewed distribution observed in each laboratory data. Additional information and graphical representations of the imputation procedure can be found in Supplement Figures 2 and 3. A comprehensive description of the missing value handling approach is provided in the "Data Exclusion and Imputation" section. 

- First, if the variable had an existing previous value, the last observation carried forward method was used (such as laboratory data and vital signs). Second, when the patient was hospitalized without laboratory results, we assumed that the prediction based on normal values until abnormal data were observed would be robust. As many data distributions were skewed, the mode was appropriate for representing normal values in our dataset. (S2,3 Figs) Thus, no previously observed data were imputed with each mode value.

- The authors should provide more information about the hyper-parameters of the models used, such as the number of hidden layers and the learning rate for the deep neural network model.

Due to space limitations in the manuscript, a detailed exposition of the parameters of the DNN (Deep Neural Network) model was omitted. The model's architecture is not overly complex, and the majority of architectural specifics have been provided in the Supplement Method section. The following provides a comprehensive description of the specific architecture utilized in the model.

- Our model is comprised of an input layer, three hidden layers, and an output layer, each utilizing a different number of nodes. The model architecture is as follows.

Input Layer: The input layer consists of nodes equal to the number of features in our training data. This layer passes the data directly to the first hidden layer.

First Hidden Layer: This layer contains 512 nodes. The layer utilizes a dropout (20%) regularization technique to prevent overfitting. The activation function employed in this layer is the Rectified Linear Unit (ReLU) function, which introduces non-linearity into the model. 

Second Hidden Layer: This layer consists of 1024 nodes. Like the first hidden layer, this layer also uses dropout regularization (20%) the ReLU activation function. 

Third Hidden Layer: This layer is made up of 512 nodes. It also applies dropout regularization and the ReLU activation function.

Output Layer: Representing the concluding layer of our model, this layer comprises three nodes. It does not incorporate dropout or an activation function. The quantity of nodes is indicative of the number of categories we aspire to forecast, which in our case are the daily severity levels: mild, moderate, and severe.

- It would be helpful to provide more information about the performance metrics used to evaluate the models.

We used the model metrics from accuracy, AUROC, AUPRC. And the importance of input variable was calculated by SHAP values. The description was described in Statistical analysis section in Method. 

- Data preprocessing and feature selection: The description of the data preprocessing and feature selection steps is clear. However, it would be useful to provide more details on the specific features selected for the models and the rationale behind their selection.

The process of feature selection was solely determined by the clinician serving as the first author. Given our extensive care of numerous COVID-19 patients, we carefully assessed patient severity based on a comprehensive set of variables, including laboratory results, vital signs, demographics, and other relevant factors. We firmly believed that these selected variables provided a sufficiently robust measure of patient severity and future outcome. As a result, we elaborated on the rationale behind the inclusion of these specific features in the "Data Processing and Feature Selection" section.

- The laboratory data encompassed a range of parameters, including complete blood cell count, chemistry markers (such as protein, albumin, liver function tests, BUN/Creatinine), electrolyte levels, inflammatory markers (CRP, LDH, ferritin, procalcitonin), and coagulation measurements. The selection of these specific variables was determined by the attending clinicians responsible for patient care, who deemed them relevant for the management and treatment of COVID-19 patients.

- The models used for prediction are well-described. However, it would be beneficial to provide more information on the hyperparameters of each model to ensure reproducibility of the study. Additionally, it would be helpful to provide references for the transformer model and clarify how it handles the temporal aspect of the data.

- As a consequence of space constraints within the manuscript, a comprehensive account of the hyperparameters for each model was relegated to the Supplement Method section. A more exhaustive and complete description of all hyperparameters can be found in the Supplement Method file.

2. The results section provides a comprehensive overview of the findings. However, it would be beneficial to include specific details on the performance metrics (AUROC and AUPRC) for each model at different prediction horizons (day 0, day 1, day 2). It would also be useful to provide statistical significance testing or confidence intervals to assess the differences in performance between models.

In order to ascertain statistical significance, we carried out random sampling of the test set 1000 times and calculated the AUROC, AUPRC, and accuracy, setting the confidence interval at 2.5% and 97.5%. The resulting data has been incorporated into Table 2. Moreover, the models’ decision based on threshold was compared using decision curve analysis. We added the result in Fig 4. You can easily compare the model performances with the likelihood of the probability and compare the model prediction in mild, moderate, and severe. 

3. The discussion provides a good interpretation of the results and highlights the importance of considering hospital-specific factors in real-time prediction models. However, it would be helpful to discuss the limitations of the study, such as the generalizability of the findings to other healthcare settings and the potential impact of missing data on the model performance. Additionally, it would be helpful to compare the findings of this study with previous studies that have predicted COVID-19 severity.

 - In this discussion, we added the potential limitation in generalizability of our model as follows. 

This study had some limitations. First, it was based on single-centered retrospective cohort data; thus, the model may not show high performance using other hospital datasets. Due to the difficulty of accessing other hospitals with a comparable patient group, external validation was restricted. 

This study had some limitations. First, it was based on single-centered retrospective cohort data; thus, the model may not show high performance using other hospital datasets. Due to the difficulty of accessing other hospitals with a comparable patient group, external validation was restricted. The variability in the frequency of vital sign and laboratory tests may result in a higher incidence of missing data. Additionally, the unavailability of daily symptom records in other hospitals could potentially present challenges to the generalizability of our model.

Minor Comments:

1. In the introduction, it would be helpful to define the term "short-term outcomes" as used in the context of COVID-19 severity prediction.

- To enhance readability, we have revised the introduction to provide a clear explanation of the short-term outcome, as depicted below.

- Prediction of future need for intensive care is required for controlling capacities for facilities in medical centers. Timely preparation of essential medical devices based on short-term forecasts (1 day or 2 days) is critical for ensuring efficient management of healthcare resources. By predicting short-term patient outcomes, informed decisions can be made regarding the potential transfer of patients to alternate hospitals or care units within the same facility's ICU.

2. In the methods section, it would be useful to provide more information on the number of patients included in each severity category (mild, moderate, severe) to assess the distribution of severity levels in the dataset.

The distribution of patients based on severity is presented in Table 1. Among the observed cases, there were 2704 patients classified as mild, 955 as moderate, and 337 as severe.

3. In the results section, it would be beneficial to include a figure or table summarizing the performance metrics (AUROC, AUPRC) of each model at different prediction horizons.

Tables presenting the outcomes for various prediction horizons have been compiled as Table 2 (utilizing one-day information) and Table 3 (employing prior hospital day information by transformer). The details regarding the information horizon employed in the transformer model have been comprehensively explicated in the Supplement Method section and Supplement Figures 6 and 7.

4. In the discussion, it would be helpful to provide some insights into the clinical implications of the findings and how real-time prediction of COVID-19 severity can improve patient care and resource allocation.

For enhanced clarity and ease of comprehension, we have included the following sentences in the discussion section.

- By integrating our model into clinical practice, clinicians can readily monitor patients who may be at risk of deterioration due to COVID-19 as well as those showing signs of recovery. Such insights prove valuable in anticipating potential strains on medical facilities and allow for proactive arrangements, including network preparations for transferring patients requiring ventilator support and intensive care unit (ICU) facilities.

5. The manuscript could benefit from proofreading and minor grammatical edits to improve readability.

- We sincerely appreciate your invaluable feedback. Rest assured that we have taken great care in addressing all minor grammatical errors and rectifying any miswritten words, thanks to the expertise of an English language specialist. While the initial revision was conducted by a native English editor, we have undertaken a meticulous review and comprehensive amendment of the entire manuscript in this subsequent revision. 

Reviewer #3: While many media sources continue to discount or obfuscate the long-term effects of COVID-19, I remain isolated to protect an immunocompromised spouse. Consequently, I am very familiar with how risk containment has changed over time and best practices. It is also cool to see a snapshot of what predicted risk from pre-pandemic to around Delta invasivity data.

For the introduction, this is an excellent summary of the first few years of the pandemic and the thesis of the manuscript. It is true that early precautionary measures were useful for reducing risk, and that this prevention or mitigation strategy was used in both mild and severe cases. I had not been aware of South Korea's efforts to minimize disease severity, but that is great. To continue reducing disease severity and better estimate hospitals being at full capacity, yes, predicting disease severity is critical. The point about event risk is well-taken. Most studies arbitrarily chose a given endpoint versus baseline, rather than taking all timepoints or dynamic timepoints into account. It is also true that many of these other studies, ours included, focused on onset of COVID-19 EHR instead of estimating recovery. Thus, predicting COVID-19 onset and recovery at the BMC is a very useful extension of prior work.

For methods, in summary, I see no problems here at all. To begin, given the secondary nature of the data, it makes sense that informed consent would be waived. The timestamped data for a robustly large cohort is also good. The data collected reflect what is available through EHRs and what is routinely collected in many tertiary care settings. Scaling variables is described surprisingly well. I commend the authors on this point. All of the decisions seem fine with regard to making variables binary, continuous, or bringing in the distribution tails (with clinician direction) when values might be unforeseen outliers. Supplementary Figures 1 and 2 also show a willingness for the raw data to be transparent, which I appreciate. The distributions are all similar to what I would expect in relatively healthy middle-aged to aged adults who present at a given clinic. Data missingness for mild cases is also understandable, as this will stochastically vary depending on the nursing staff, attending physician, and capacity of the tertiary clinic. Supplemental Table 1 goes into this in detail. For imputation, it is reasonable to include the mode to avoid bizarre behavior for vitals and other data in estimation analyses. I appreciate the additional data in Supplement Figures 3-5 that describe raw data, as well as data fit for mild to moderate COVID-19 cases using different estimation methods. For statistical methods, this all seems standard regarding classification metrics, split-model training vs. assessment probands, and even some reasonable mean +/- SD for input length.

For results, it is refreshing and welcome to read a brief summary of Table 1, as well as initial figures, and for it all to make intuitive sense. Table 2 reveals that regardless of the model type tested, the AUROC or AUPRC was outstanding. It was interesting how RF did better than DNN. Given the sparsity of the model set and N, however, too many interaction terms may have loaded and diluted overall model fit.

My only suggestion here is to list, either in text or the tables, if Model X significantly differed from Model Y (e.g., if Prediction Horizon day 0, 1, or 2 showed any difference for Input Length for Accuracy). In other words, just some basic statistics to formally show what is described in section 3.1. In section 3.2, the authors describe an intriguing pattern in the data: that the best fit methods predominantly extracted hospital/treatment factors (i.e., external factors) compared to DNN which extracted patient factors (i.e., internal factors). To strengthen or formalize this observation, a statistical test comparing factors on a binary scale ('0' = internal, '1' = external) might be useful. This is just a friendly suggestion for substantiating the claim made and is not a critique.

For the discussion, I again agree that predicting recovery is just as important as initial infection and degree of disease aggravation. Comparisons with other studies are appropriate and thoughtful. The strengths and limitations sections are both thorough and, again, thoughtful.

- Thank you for your exceptional review. Your thoughtful feedback has been immensely valuable, and we genuinely appreciate the insightful comments you have provided. While we recognized the differences in variable characteristics, we found it challenging to effectively illustrate these distinctions in our analysis. We eagerly welcome your valuable insights in this regard.

- To address the differentiation in input variables, we have incorporated a classification that categorizes variables as either extrinsic (hospital-related) or intrinsic (patient-related) in the Statistical Analysis section. 

Input variable importance was assessed based on the data type, distinguishing between hospital/treatment-dependent factors (extrinsic factors) and patient-dependent factors (intrinsic factors). Hospital/treatment-dependent factors encompassed the number of vital sign occurrences (count, variance) and the dosage of drugs (max, min, median). On the other hand, patient-dependent factors included vital sign values (max, mean, median, min), demographics, and the presence of symptoms.

- And the proportion of intrinsic factor in top 20 importance variable was described in result section as below. Despite the lack of clear distinction based on the chi-square p-value in this classification (intrinsic vs extrinsic), we believe that the issue lies with the number of observations rather than actual differences between the models. Consequently, we posit that tree-based models may carry a higher risk of bias for time-series analysis.

- The factors were categorized as patient-dependent and hospital/treatment-dependent. For predicting patient’s severity, the patient factor cannot be altered by the hospital where the patient was admitted. However, hospital/treatment factors, such as the number of vital signs that should be monitored and the dosage of steroid administered, can differ between hospitals. Regarding the SHAP variable analysis, the patient-dependent percentage for each model was 95% in DNN, 75% in XGBoost, and 35% in random forest. (p-value: 0.065).

---

## [Decision Letter · Decision Letter 1]

23 Oct 2023

PONE-D-23-14493R1In-hospital real-time prediction of COVID-19 severity regardless of disease phase using electronic health recordsPLOS ONE

Dear Dr. Jeong,

Thank you for submitting your manuscript to PLOS ONE. After careful consideration, we feel that it has merit but does not fully meet PLOS ONE’s publication criteria as it currently stands. Therefore, we invite you to submit a revised version of the manuscript that addresses the points raised during the review process.

We look forward to receiving your revised manuscript.

Kind regards,

John Adeoye

Academic Editor

PLOS ONE

Journal Requirements:

Reviewers' comments:

Reviewer's Responses to Questions

**Comments to the Author**

1. If the authors have adequately addressed your comments raised in a previous round of review and you feel that this manuscript is now acceptable for publication, you may indicate that here to bypass the “Comments to the Author” section, enter your conflict of interest statement in the “Confidential to Editor” section, and submit your "Accept" recommendation.

Reviewer #3: All comments have been addressed

Reviewer #4: All comments have been addressed

2. Is the manuscript technically sound, and do the data support the conclusions?

Reviewer #3: Yes

Reviewer #4: Yes

3. Has the statistical analysis been performed appropriately and rigorously? 

Reviewer #3: Yes

Reviewer #4: Yes

4. Have the authors made all data underlying the findings in their manuscript fully available?

Reviewer #3: Yes

Reviewer #4: Yes

5. Is the manuscript presented in an intelligible fashion and written in standard English?

Reviewer #3: Yes

Reviewer #4: Yes

6. Review Comments to the Author

Reviewer #3: The authors have addressed my comments. I have no further concerns. I think the authors did a good job addressing my comments

Reviewer #4: The article discusses the development and evaluation of machine learning and deep learning models to predict the daily severity of COVID-19 in patients at a dedicated hospital. The goal is to forecast the severity of the condition up to 2 days in advance to help manage healthcare resources efficiently. The study uses a dataset of COVID-19 patients from a specific medical center and assesses various models for their predictive accuracy.

The key findings and points in the article include:

- The importance of predicting the daily severity of COVID-19 to facilitate proactive resource allocation, such as respiratory devices.

- The comparison of different types of prediction models, including non-temporal (logistic regression, random forest, gradient boost) and temporal models (transformer).

- The use of patient data, including demographics, symptoms, laboratory tests, and vital signs as input features for prediction.

- The performance of the models in predicting severity for different time horizons (day 0, day 1, and day 2), with the random forest model outperforming others for predicting day 0 severity.

- The importance of feature selection and model interpretability using the SHAP method, with differences in feature importance between models.

- The discussion of model generalizability to other hospitals and the potential limitations of the study, including data availability and clinical variations.

In conclusion, the study presents a machine learning model, particularly the random forest model, as a valuable tool for predicting COVID-19 severity, aiding healthcare resource allocation, and offering insights into patient outcomes. The hierarchical transformer model also demonstrated good performance for certain periods of hospital admission. Further validation and research are needed to confirm the findings and adapt the model for use in different healthcare settings.

Comments that should be addressed by the authors:

1. Early identification of patients who are on the path to recovery could provide a justification for reducing or discontinuing ventilatory support and potentially adjusting the timing of treatment interventions?

2. The utilization of an exceptionally large number of predictors raises concerns about potential overfitting of the model to the dataset.

3. The algorithm predicts severity of disease within 2 days. How should a clinician interpret the results of the algorithm if the prediction is opposed to clinical evidence? For example, if a patient is at clinical stability and the algorithm predicts impeding severe disease, what kind of information can be deducted by the clinician to address the causes of impending aggravation of the disease?

7. PLOS authors have the option to publish the peer review history of their article (what does this mean?). If published, this will include your full peer review and any attached files.

Reviewer #3: **Yes: **Auriel A. Willette

Reviewer #4: No

---

## [Author Response · Author response to Decision Letter 1]

24 Oct 2023

Review Comments to the Author

Reviewer #3: The authors have addressed my comments. I have no further concerns. I think the authors did a good job addressing my comments

- Thank you for your positive feedback and acknowledgment of our efforts in addressing your comments. We greatly appreciate your constructive insights throughout the review process.

Reviewer #4: The article discusses the development and evaluation of machine learning and deep learning models to predict the daily severity of COVID-19 in patients at a dedicated hospital. The goal is to forecast the severity of the condition up to 2 days in advance to help manage healthcare resources efficiently. The study uses a dataset of COVID-19 patients from a specific medical center and assesses various models for their predictive accuracy.

The key findings and points in the article include:

- The importance of predicting the daily severity of COVID-19 to facilitate proactive resource allocation, such as respiratory devices.

- The comparison of different types of prediction models, including non-temporal (logistic regression, random forest, gradient boost) and temporal models (transformer).

- The use of patient data, including demographics, symptoms, laboratory tests, and vital signs as input features for prediction.

- The performance of the models in predicting severity for different time horizons (day 0, day 1, and day 2), with the random forest model outperforming others for predicting day 0 severity.

- The importance of feature selection and model interpretability using the SHAP method, with differences in feature importance between models.

- The discussion of model generalizability to other hospitals and the potential limitations of the study, including data availability and clinical variations.

In conclusion, the study presents a machine learning model, particularly the random forest model, as a valuable tool for predicting COVID-19 severity, aiding healthcare resource allocation, and offering insights into patient outcomes. The hierarchical transformer model also demonstrated good performance for certain periods of hospital admission. Further validation and research are needed to confirm the findings and adapt the model for use in different healthcare settings.

Comments that should be addressed by the authors:

1. Early identification of patients who are on the path to recovery could provide a justification for reducing or discontinuing ventilatory support and potentially adjusting the timing of treatment interventions?

 - We value your insights. In our study, the primary goal was to proactively detect signs of patient deterioration and recovery. Although our model doesn't directly change patient trajectory upon early identification, its strength lies in the anticipation of potential deterioration. This early prediction becomes pivotal when considering resource allocation. On days where the model predicts fewer severe cases, receiving a call about a deteriorating patient might lead to more flexible allocation of resources, such as quickly assigning ventilatory support like HFNC. However, on days with a higher predicted count of at-risk patients, the same call might necessitate more judicious and possibly stringent resource allocation, ensuring that the most critical patients receive timely intervention. This nuanced approach to resource allocation, guided by our predictive model, aims to optimize patient care outcomes in varied scenarios.

2. The utilization of an exceptionally large number of predictors raises concerns about potential overfitting of the model to the dataset.

 We acknowledge your concern regarding the potential for overfitting due to the extensive number of predictors. It's worth noting that our inputs predominantly consist of variables routinely collected during standard patient care, enhancing their potential for generalization across different hospitals. While many contemporary predictive models strive to incorporate as many inputs as possible, we are aware of the risks associated with overfitting. To mitigate this, we employed a methodological approach by partitioning the dataset into training, validation, and test sets—a standard practice that we have applied in our study. Furthermore, we recognize the potential for bias, especially given that some high-performing tree-based models may produce overfitted results by prioritizing hospital-dependent factors. As a proactive measure, we advocate for the use of deep learning models that exclude these potential biases (hospital-dependent elements), ensuring not only effective near-term outcome prediction but also facilitating model generalization across various hospitals. 

3. The algorithm predicts severity of disease within 2 days. How should a clinician interpret the results of the algorithm if the prediction is opposed to clinical evidence? For example, if a patient is at clinical stability and the algorithm predicts impeding severe disease, what kind of information can be deducted by the clinician to address the causes of impending aggravation of the disease?

We recognize the significance of your concern. In time-series prediction models, including ours, there's an inherent challenge when the model's forecast diverges from the observed clinical status. This challenge isn't unique to our model but is a general consideration for all time-series prediction models. In our current study, our primary focus has been on forecasting potential risks and recovery. The question of how to act upon a prediction, especially when it appears to contradict clinical observations, is indeed the subsequent phase of our model's objectives. We understand that further prospective studies are warranted to assess the potential risks associated with such discrepancies. For the time being, our model has been developed with an emphasis on addressing the challenges of overcrowded hospitals, where clinicians are constantly grappling with decisions related to ICU admissions, HFNC assignments, and potential discontinuations. In such scenarios, our model could serve as a valuable tool for healthcare professionals navigating high-demand environments.

---

## [Editor Report · Decision Letter 2]

26 Oct 2023

PONE-D-23-14493R2In-hospital real-time prediction of COVID-19 severity regardless of disease phase using electronic health recordsPLOS ONE

Dear Dr. Jeong,

Thank you for submitting your manuscript to PLOS ONE. After careful consideration, we feel that it has merit but does not fully meet PLOS ONE’s publication criteria as it currently stands. Therefore, we invite you to submit a revised version of the manuscript that addresses the points raised during the review process.

We look forward to receiving your revised manuscript.

Kind regards,

John Adeoye

Academic Editor

PLOS ONE

Journal Requirements:

**Additional Editor Comments:**

Dear Authors,

Please include the concerns raised by 'Reviewer #4' in the last review round as potential limitations or recommendations for future work in the manuscript.

---

## [Author Response · Author response to Decision Letter 2]

29 Oct 2023

Response to Reviewer

Reviewer #3: The authors have addressed my comments. I have no further concerns. I think the authors did a good job addressing my comments

- Thank you for your positive feedback and acknowledgment of our efforts in addressing your comments. We greatly appreciate your constructive insights throughout the review process.

Reviewer #4: The article discusses the development and evaluation of machine learning and deep learning models to predict the daily severity of COVID-19 in patients at a dedicated hospital. The goal is to forecast the severity of the condition up to 2 days in advance to help manage healthcare resources efficiently. The study uses a dataset of COVID-19 patients from a specific medical center and assesses various models for their predictive accuracy.

The key findings and points in the article include:

- The importance of predicting the daily severity of COVID-19 to facilitate proactive resource allocation, such as respiratory devices.

- The comparison of different types of prediction models, including non-temporal (logistic regression, random forest, gradient boost) and temporal models (transformer).

- The use of patient data, including demographics, symptoms, laboratory tests, and vital signs as input features for prediction.

- The performance of the models in predicting severity for different time horizons (day 0, day 1, and day 2), with the random forest model outperforming others for predicting day 0 severity.

- The importance of feature selection and model interpretability using the SHAP method, with differences in feature importance between models.

- The discussion of model generalizability to other hospitals and the potential limitations of the study, including data availability and clinical variations.

In conclusion, the study presents a machine learning model, particularly the random forest model, as a valuable tool for predicting COVID-19 severity, aiding healthcare resource allocation, and offering insights into patient outcomes. The hierarchical transformer model also demonstrated good performance for certain periods of hospital admission. Further validation and research are needed to confirm the findings and adapt the model for use in different healthcare settings.

Comments that should be addressed by the authors:

1. Early identification of patients who are on the path to recovery could provide a justification for reducing or discontinuing ventilatory support and potentially adjusting the timing of treatment interventions?

 - We value your insights. In our study, the primary goal was to proactively detect signs of patient deterioration and recovery. Although our model doesn't directly change patient trajectory upon early identification, its strength lies in the anticipation of potential deterioration. This early prediction becomes pivotal when considering resource allocation. On days where the model predicts fewer severe cases, receiving a call about a deteriorating patient might lead to more flexible allocation of resources, such as quickly assigning ventilatory support like HFNC. However, on days with a higher predicted count of at-risk patients, the same call might necessitate more judicious and possibly stringent resource allocation, ensuring that the most critical patients receive timely intervention. This nuanced approach to resource allocation, guided by our predictive model, aims to optimize patient care outcomes in varied scenarios.

2. The utilization of an exceptionally large number of predictors raises concerns about potential overfitting of the model to the dataset.

 We acknowledge your concern regarding the potential for overfitting due to the extensive number of predictors. It's worth noting that our inputs predominantly consist of variables routinely collected during standard patient care, enhancing their potential for generalization across different hospitals. While many contemporary predictive models strive to incorporate as many inputs as possible, we are aware of the risks associated with overfitting. To mitigate this, we employed a methodological approach by partitioning the dataset into training, validation, and test sets—a standard practice that we have applied in our study. Furthermore, we recognize the potential for bias, especially given that some high-performing tree-based models may produce overfitted results by prioritizing hospital-dependent factors. As a proactive measure, we advocate for the use of deep learning models that exclude these potential biases (hospital-dependent elements), ensuring not only effective near-term outcome prediction but also facilitating model generalization across various hospitals. 

3. The algorithm predicts severity of disease within 2 days. How should a clinician interpret the results of the algorithm if the prediction is opposed to clinical evidence? For example, if a patient is at clinical stability and the algorithm predicts impeding severe disease, what kind of information can be deducted by the clinician to address the causes of impending aggravation of the disease?

We recognize the significance of your concern. In time-series prediction models, including ours, there's an inherent challenge when the model's forecast diverges from the observed clinical status. This challenge isn't unique to our model but is a general consideration for all time-series prediction models. In our current study, our primary focus has been on forecasting potential risks and recovery. The question of how to act upon a prediction, especially when it appears to contradict clinical observations, is indeed the subsequent phase of our model's objectives. We understand that further prospective studies are warranted to assess the potential risks associated with such discrepancies. For the time being, our model has been developed with an emphasis on addressing the challenges of overcrowded hospitals, where clinicians are constantly grappling with decisions related to ICU admissions, HFNC assignments, and potential discontinuations. In such scenarios, our model could serve as a valuable tool for healthcare professionals navigating high-demand environments.

---

## [Editor Report · Decision Letter 3]

31 Oct 2023

In-hospital real-time prediction of COVID-19 severity regardless of disease phase using electronic health records

PONE-D-23-14493R3

Dear Dr. Jeong,

We’re pleased to inform you that your manuscript has been judged scientifically suitable for publication and will be formally accepted for publication once it meets all outstanding technical requirements.

Kind regards,

John Adeoye

Academic Editor

PLOS ONE
---

## [Editor Report · Acceptance letter]

18 Jan 2024

PONE-D-23-14493R3 

PLOS ONE

Dear Dr. Jeong, 

I'm pleased to inform you that your manuscript has been deemed suitable for publication in PLOS ONE. Congratulations! Your manuscript is now being handed over to our production team.

Kind regards, 

on behalf of

Dr. John Adeoye 

Academic Editor

PLOS ONE